# Estimation of Soil Moisture Content Based on Fractional Differential and Optimal Spectral Index

Wangyang Li, Youzhen Xiang *, Xiaochi Liu, Zijun Tang, Xin Wang, Xiangyang Huang, Hongzhao Shi, Mingjie Chen, Yujie Duan, Liaoyuan Ma, Shiyun Wang, Yifang Zhao, Zhijun Li and Fucang Zhang

The Key Laboratory of Agricultural Soil and Water Engineering in Arid Areas, Ministry of China, Northwest A&F University, Xianyang 712100, China; lwy1222@nwafu.edu.cn (W.L.); 2023055903@nwsuaf.edu.cn (X.L.); tangzijun@nwsuaf.edu.cn (Z.T.); wx123@nwsuaf.edu.cn (X.W.); 2023055900@nwsuaf.edu.cn (X.H.); shihongzhao7@nwafu.edu.cn (H.S.); 2023012631@nwafu.edu.cn (M.C.); 2023012614@nwafu.edu.cn (Y.D.); 2023012608@nwafu.edu.cn (L.M.); wsydyxhs2023@nwafu.edu.cn (S.W.); 2023012616zyf@nwafu.edu.cn (Y.Z.); lizhij@nwsuaf.edu.cn (Z.L.); zhangfc@nwsuaf.edu.cn (F.Z.)
* Correspondence: youzhenxiang@nwsuaf.edu.cn

**Abstract:** Applying hyperspectral remote sensing technology to the prediction of soil moisture content (SMC) during the growth stage of soybean emerges as an effective approach, imperative for advancing the development of modern precision agriculture. This investigation focuses on SMC during the flowering stage under varying nitrogen application levels and film mulching treatments. The soybean canopy's original hyperspectral data, acquired at the flowering stage, underwent 0–2-order differential transformation (with a step size of 0.5). Five spectral indices exhibiting the highest correlation with SMC were identified as optimal inputs. Three machine learning methods, namely support vector machine (SVM), random forest (RF), and back propagation neural network (BPNN), were employed to formulate the SMC prediction model. The results indicate the following: (1) The correlation between the optimal spectral index of each order, obtained after fractional differential transformation, and SMC significantly improved compared to the original hyperspectral reflectance data. The average correlation coefficient between each spectral index and SMC under the 1.5-order treatment was 0.380% higher than that of the original spectral index, with mNDI showing the highest correlation coefficient at 0.766. (2) In instances of utilizing the same modeling method with different input variables, the SMC prediction model's accuracy follows the order: 1.5 order > 2.0 order > 1.0 order > 0.5 order > original order. Conversely, with consistent input variables and a change in the modeling method, the accuracy order becomes RF > SVM > BPNN. When comprehensively assessing model evaluation indicators, the 1.5-order differential method and RF method emerge as the preferred order differential method and model construction method, respectively. The $R^2$ for the optimal SMC estimation model in the modeling set and validation set were 0.912 and 0.792, RMSEs were 0.005 and 0.004, and MREs were 2.390% and 2.380%, respectively. This study lays the groundwork for future applications of hyperspectral remote sensing technology in developing soil moisture content estimation models for various crop growth stages and sparks discussions on enhancing the accuracy of these different soil moisture content estimation models.

**Keywords:** soybean; hyperspectrum; fractional order differentiation; optimal spectral index; soil moisture content

## 1. Introduction

Soybean is one of the most essential oil products in the world agricultural trade [1]. China, a significant soybean producer, has consistently ranked fourth globally in total output in recent years [2]. In the past, China was one of the major consumers of soybean; China used to purchase around 62% of the soybean that was traded internationally, and only 14.3% of its soybean consumption was self-sufficient. The self-sufficiency rate of soybean is less than 15%, which strains the global food supply [3]. The production and

quality of soybeans hold a crucial strategic role in constructing a modern product with distinct Chinese characteristics. Soybean output and quality directly impact China's food security level [4]. The developmental condition of soybeans during the flowering stage significantly affects their subsequent reproductive growth and ultimate yield formation. A soil moisture content (SMC) that is too high or too low will directly affect the flowering quality of soybean, thus affecting the yield and quality of soybean [5,6]. A scientific and efficient acquisition of SMC is significant for growth status evaluation and yield prediction of soybean at the flowering stage [7].

SMC plays a crucial role in the creation, alteration, and utilization of surface water resources [8]. SMC stands as a crucial indicator for assessing crop soil drought conditions, while also serving as a pivotal determinant of crop yield and overall crop quality [9]. Monitoring soil moisture holds substantial significance in advancing water-efficient irrigation practices and optimizing water resource utilization [5]. Many conventional drying methods of determining SMC persist, as the drying method is characterized by its high precision and accuracy [10]. Notably, it has been formally adopted as both a national and international standard. Nevertheless, in situ methodologies for measuring SMC face difficulties in consistently attaining continuous observations and require considerable time and labor [11]. Zai Songmei's research has revealed the feasibility of using a spectrophotometer to determine soil moisture content, providing evidence for the rapid assessment of soil moisture content through soil spectral characteristics [12]. Cao Qi's study found that the radar ground wave method can accurately invert soil volumetric water content, but inversion accuracy is influenced by land use types [13]. Gamma-ray transmission methods have been precisely employed in the study of agricultural soil properties [14]. A study by Medhat discovered that the gamma-ray transmission method, utilizing a portable cadmium telluride detector, offers practical, cost-effective, nondestructive, and rapid analytical advantages in measuring soil density and volumetric water content. However, it places high demands on instrument accuracy [15,16].

Researchers have utilized remote sensing data to explore the intricate relationship between soil spectral properties and moisture levels [17]. However, the direct evaluation of soil moisture through remote sensing technology necessitates extensive field sampling periods. Hence, the trend towards indirect SMC detection, established by relating crop reflectance to soil moisture, is on the rise. The direct link between reflectance and vegetation water content forms the cornerstone of SMC monitoring techniques [18]. Numerous studies have underscored the reflectance–water content relationship across various crops, wheat included [19–22]. Additionally, Gouvea et al. [23] probed the impact of soil water stress on physiological traits like photosynthesis, stomatal conductance, transpiration, and $CO_2$ levels in soybean plants. Thus, the utilization of remote sensing technology and its correlation with crop canopy reflectance provides a viable avenue for SMC monitoring.

While numerous studies have investigated the derivation of crop water status via spectral measurements, the direct assessment of SMC through crop reflectance remains relatively rare. Sobrino et al. [24] estimated soil moisture content across diverse crops and growth stages using both airborne hyperspectral scanners (AHSs) and satellite imagery. Panigrahi and Das employed ground-based hyperspectral measurements to simulate soil water potential in paddy fields during multiple growth stages [25]. However, the effectiveness of soil moisture prediction models utilizing canopy spectral data can be impeded by factors such as canopy structure, leaf area, angles, positions, shadows, and soil backgrounds [26,27]. These complexities hinder the establishment of a robust quantitative relationship between soil moisture and canopy reflectance, consequently limiting the generalizability of developed models when applied to novel agricultural regions [28].

Certain researchers have leveraged the integer differential transformation method to preprocess raw hyperspectral reflectance data, thus attenuating background noise to some degree and bolstering modeling precision [29]. Nevertheless, it is worth noting that the application of first-order, second-order, and even higher-order integer-order differential transformation techniques, while dampening background noise, has been observed to

disregard the continuity and gradient of spectral information, ultimately leading to a loss of spectral characteristics [30,31]. This scenario prompts the emergence of fractional differential transform, a mathematical extension of the integer-order differential approach. This novel approach holds the capability to accentuate subtle shifts within spectral information, enhancing weaker spectral features and amplifying the signal-to-noise ratio inherent in spectral reflectance [32]. Tang et al. [33] demonstrated the estimation of soil salinity through a machine learning framework grounded in remote sensing fractional derivative. Despite these advancements, limited research has been dedicated to investigating the nuanced connection between crop canopy reflectance and soil moisture subsequent to fractional differential transformation.

This study investigates the flowering stage SMC across various nitrogen application rates and film mulching treatments. The original hyperspectral reflectance data underwent fractional differential transformation of 0–2 orders with a step size of 0.5. Employing the correlation matrix method, spectral bands exhibiting the highest correlation with SMC within the 350 to 1830 nm range were identified, leading to the construction of five sets of 25 optimal spectral indices. Building upon this foundation, a predictive model for flowering SMC was established using support vector machine (SVM), random forest (RF), and back propagation neural network (BPNN) algorithms. This study further scrutinized the influence of diverse differential orders and machine learning methods on the predictive accuracy of the SMC model. By doing so, this investigation aims to furnish a theoretical framework that contributes to a more precise and rapid determination of flowering SMC. It also provides a theoretical basis for artificial intelligence applications to quickly and accurately analyze large-scale hyperspectral satellite remote sensing information [34].

## 2. Materials and Methods

### 2.1. Research Area and Test Design

In this study, a two-year (2021–2022) soybean field experiment was conducted at the Institute of Water-Saving Agriculture (34°18′ N, 108°24′ E, 524.7 m a.s.l.) within the Key Laboratory of Agricultural Water and Soil Engineering of the Ministry of Education at Northwest A & F University (Figure 1). The mean maximum temperature for June to October 2021 was 30.3 °C, with a minimum of 20.0 °C, while in 2022, it was 31.3 °C and 21.2 °C, respectively. Precipitation during the sowing period in 2021 and 2022 was 432.6 mm and 279.5 mm, respectively. In comparison to the 30-year average rainfall of 345 mm during the soybean season (1991–2020), 2021 was categorized as a wet year, whereas 2022 was considered a dry year.

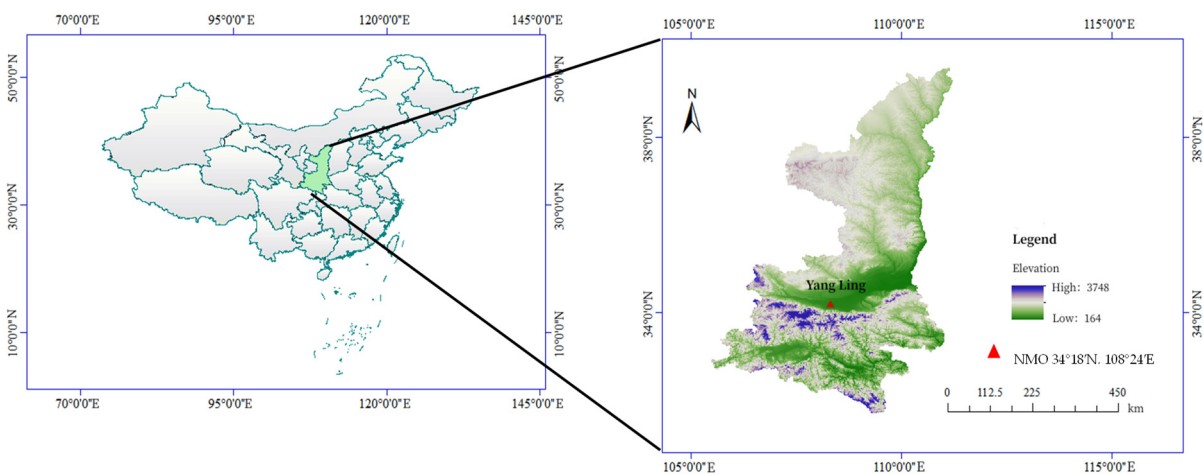

**Figure 1.** Study area.

In this experiment, three nitrogen application levels were set: N1: 60 kg/hm$^2$, N2: 120 kg/hm$^2$, N3: 180 kg/hm$^2$. Four mulching treatments were also set: NM: no mulching

treatment, SM: straw mulching, FM: agricultural film mulching, SFM: straw + agricultural film mulching. The straw mulching amount was 6000 kg/hm$^2$. The FM and SFM treatments were carried out by ridging and covering the film side. Two ridges with a width of 40 cm and a height of 25 cm were raised in the experimental plot, and the ridge surface was covered with a 60 cm wide plastic film. Soybeans were sown at 5 cm on the film side with a row spacing of 50 cm. This study designed 12 treatments, 2 replicates, and a total of 24 plots. The experimental plot area was 2.5 m × 6 m = 15 m$^2$. Slow-release nitrogen fertilizer (SNF), potassium fertilizer (K$_2$O, 30 kg/hm$^2$), and phosphorus fertilizer (P$_2$O$_5$, 30 kg/hm$^2$) were applied as base fertilizer before sowing. The soybean variety used in the experiment was Shanning 17. It was sown on 17 June 2021 and 9 June 2022 and harvested on 30 September 2021 and 28 September 2022, respectively. The soybean flowering periods were 28 July 2021–24 August 2021 and 23 July 2022–20 August 2022, respectively. There were no obvious diseases and insect pests during the soybean growth period. The experimental scheme design of this study is shown in Table 1.

**Table 1.** Test scheme design.

| SFM | NM | SFM | SM | SM | FM |
|-----|-----|-----|-----|-----|-----|
| N1 | N3 | N2 | N3 | N2 | N1 |
| NM | SFM | SM | FM | FM | NM |
| N1 | N3 | N1 | N3 | N2 | N1 |
| SFM | SFM | SFM | NM | NM | FM |
| N2 | N1 | N3 | N3 | N2 | N3 |
| SM | SM | SM | NM | FM | FM |
| N2 | N1 | N3 | N2 | N2 | N1 |

*2.2. Measurements and Methods*

2.2.1. Data Acquisition

This study used the most basic drying method to measure SMC during the soybean flowering period (6 August 2021 and 10 August 2022) in the two-year experiment. At the same time, the spectral data were collected. The weather was sunny, and the light was stable. During the flowering period of soybean, soil samples were taken from the middle of two plants, at a position of 15 cm from each plant and at a horizontal position in the middle of bare land. Soil samples were taken at intervals of 20 cm in each soil layer from 0 cm to 60 cm of soil depth. The soil samples were mixed thoroughly and dried in an oven at 105 °C for 8 h to determine SMC. In this study, six sites were randomly selected in each plot during the flowering period of soybean to determine soil moisture content. The mean value of the six sampling points was the soil moisture content of the plot for a total of 24 plots, and the corresponding hyperspectral remote sensing information was obtained at the same time. A total of 48 groups of SMC and hyperspectral reflectance samples were obtained and tested in the two-year experiment (Table 2).

**Table 2.** Statistics of soil moisture content of soybean at flowering stage.

| Statistical Indicators | Soil Moisture Content |
|------------------------|-----------------------|
| Sample Size | 48.00 |
| Maximum Value | 0.16 |
| Minimum Value | 0.11 |
| Mean Value | 0.08 |
| Standard Deviation | 0.01 |
| Coefficient of Variation/% | 12.5 |

This study selected the original hyperspectral information under the SFMN3, SMN3, FMN3, and NMN3 treatments and plotted its spectral characteristic curves, as shown in Figure 2.

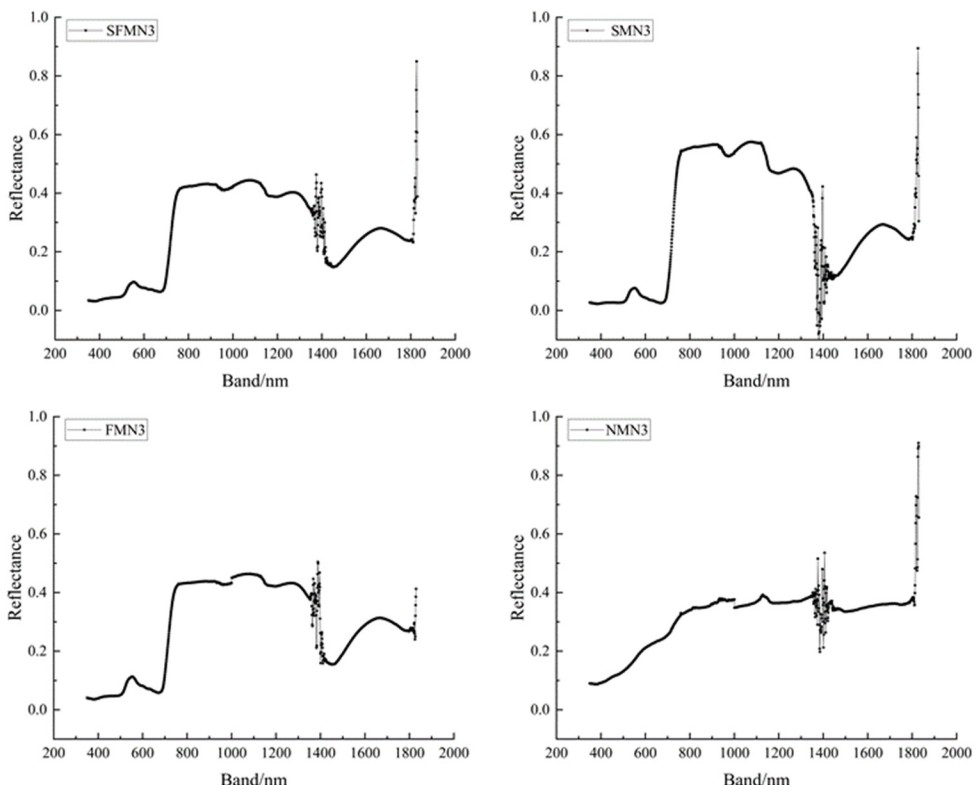

**Figure 2.** Hyperspectral reflectance characteristic curve graph.

### 2.2.2. Spectral Data Acquisition of Soybean Crown Height

This study collected the hyperspectral reflectance data of soybean canopy in the experimental area from 10:40 a.m. to 1:00 p.m. on 6 August 2021 and 10 August 2022, respectively. During this period, the light was sufficient, the spectral information was measured, and the error was small. The hyperspectral measurement instrument for the experiment was the Field Spec4 visibl/near-infrared portable ground object hyperspectral spectrometer produced by the American company ASD (Analytical Spectral Devices, Inc., Boulder, CO, USA). The instrument's band range is 350~1830 nm. The spectral resolution of 350~1000 nm is 3 nm, and the sampling interval is 1.4 nm. The 1000~1830 nm resolution is 10 nm, and the sampling interval is 2 nm. The instrument automatically interpolates the sampling data to 1 nm interval output, and the field of view is 25°.

Prior to obtaining hyperspectral data, the spectrometer underwent preheating and optimization, with the reference plate test and comparison completed within 1 min. Following the acquisition of hyperspectral reflectance data for the initial test area, reference plate correction was conducted before acquiring hyperspectral reflectance data for the subsequent test area. In each experimental plot, a crop canopy with balanced growth was selected. The testers held a spectral sensor probe and collected data vertically downwards 75 cm from the top of the crop canopy. A total of 10 hyperspectral reflectance data were collected and recorded each time. According to the '3σ' principle [33], each plot's final hyperspectral reflectance data were used as the average value of the remaining spectral bands.

### 2.2.3. Soil Moisture Content Data Collection

In this study, the most basic drying method was used to determine the SMC in the field. Three sites were randomly selected in 48 experimental plots to collect soil samples. After

drying, the mean value was calculated as the measured value of each experimental field, and the data were collected on the day of hyperspectral reflectance data image acquisition.

*2.3. Spectral Data Processing*

In this study, the Savitzky–Golay (SG) smoothing method was used to denoise the spectral data. The G-L fractional differential algorithm executed a 0–2-order (step size 0.5) fractional differential transformation on the hyperspectral reflectance data. This algorithm has the capability to extend the conventional integer order differential to any order differential, enabling a more comprehensive representation of nuanced changes and overall data information. The formula for the *α*-order differential of hyperspectral reflectance data in this experiment is as follows [29]:

$$\frac{d^{\alpha} f(x)}{dx^{\alpha}} \approx f(x) + (-\alpha)f(x-1) + \frac{(-\alpha)(-\alpha+1)}{2}f(x-2) + ... + \frac{\Gamma(-\alpha+1)}{n!\Gamma(-\alpha+n1)}f(x-n) \quad (1)$$

In the formula, $x$ is the value of the corresponding point; $\alpha$ is the fractional differential order; $\Gamma$ is the Gamma function; and $n$ is the difference between the upper and lower limits of differentiation. The order is 0, indicating no preprocessing.

Spectral data preprocessing and spectral index calculation were completed in MAT-LAB2021 (MathWorks, Inc., Natick, MA, USA), and Origin2021 (Origin Lab Corp., Northampton, MA, USA) was used for graphic drawing.

*2.4. Selection and Construction of Spectral Index*

In this study, 10 spectral indices were selected. The ratio vegetation index (RI) and triangular vegetation index (TVI) have a strong correlation with chlorophyll content and LAI of plants, but when vegetation is dense, their sensitivity will be reduced [33]. The modified simple ratio (mSR) and modified normalized difference index (mNDI) can optimize the specular emission effect of leaves and are sensitive to changes in leaves [35]. The difference vegetation index (DI), normalized difference vegetation index (NDVI), and soil-adjusted vegetation index (SAVI) can reflect the background influence of the plant canopy and eliminate some radiation errors [36,37]. The correlation between the three-band index (TBI-1, TBI-2, TBI-3) and SMC is more stable [38]. The specific spectral index formula is shown in Table 3.

**Table 3.** Spectral index and construction formula.

| Select Index | Computing Formula | Reference |
|:---:|:---:|:---:|
| RI | $R_i/R_j$ | [33] |
| TVI | $0.5\left[120(R_i - R_{550}) - 200\left(R_j - R_{550}\right)\right]$ | [33] |
| DI | $R_i - R_j$ | [36] |
| NDVI | $R_i - R_j/R_i + R_j$ | [36] |
| SAVI | $(1+0.16)\frac{R_i - R_j}{R_i + R_j + 0.16}$ | [37] |
| mSR | $R_i - R_{455}/R_j - R_{455}$ | [35] |
| mNDI | $R_i - R_i/R_j + R_j - 2R_{455}$ | [35] |
| TBI-1 | $R_{1400}/\left(R_i + R_j\right)$ | [38] |
| TBI-2 | $\left(R_{1400} - R_i\right)/\left(R_{1400} + R_j\right)$ | [38] |
| TBI-3 | $\left(R_{1400} - R_i\right) - \left(R_{\lambda 1} - R_j\right)$ | [38] |

$R_i$ ($i$ = 1, 2, 3) is reflectance at any band; $R_j$ ($j$ = 1, 2, 3) is reflectance at any band; $R_{455}$ and $R_{1400}$ are hyperspectral reflectance at 455 nm and 1400 nm wavelengths.

*2.5. Model Construction*

In this research study, the most effective combination of spectral indices of various orders served as the input variable, and three machine learning techniques, namely BPNN, SVM, and RF, were employed for modeling and predicting SMC during the flowering stage.

The training-set-to-validation-set ratio was set at 2:1, and the final model fitting result in this experiment was determined by averaging multiple prediction fitting outcomes from the machine learning models.

BPNN is a multi-layer network that propagates forward according to error and is mostly used to solve difficult nonlinear problems [39]. The optimal BPNN combination forecasting model uses m prediction methods to obtain the predicted results as the input of the network and the actual historical data value as the network's output. The weights of various prediction methods in the prediction are obtained according to the self-learning of the network [40].

SVM is a binary classification machine learning algorithm utilizing a Gaussian kernel and polynomial kernel as the foundational kernel function. The weight coefficient is optimized using a gradient descent algorithm. SVM demonstrates an excellent generalization ability and robustness, without the issue of overfitting. Its widespread applications include pattern recognition, classification, and small-sample regression analysis, guided by the principle of minimizing cross-validation error [41,42].

RF is a composite model founded on the 'Bagging' model. Due to its simplicity and convenience, it finds extensive application in diverse regression and prediction challenges. As the RF model employs weighted averages of each tree's results to attain the final output, its implementation involves constructing numerous decision trees. The model builds a set of decision trees through the exchange and alteration of covariates to enhance prediction performance [43,44]. This study determined the number of decision trees in the RF model to be 100 after multiple training and error analyses.

*2.6. Data Processing*

(1)   Model evaluation index

The model fitting results were evaluated by a determination coefficient ($R^2$), root mean square error (RMSE), and mean relative error (MRE) [45,46]. The closer $R^2$ is to 1, the higher the model's prediction accuracy is. The smaller the MRE, the more stable the performance of the model and the more concentrated the prediction results. The calculation formula is as follows:

$$R^2 = \frac{\sum_{i=1}^{n}(\hat{y}_i - \overline{y})^2}{\sum_{i=1}^{n}(y_i - \overline{y})^2} \tag{2}$$

$$\text{RMSE} = \sqrt{\frac{\sum_{i=1}^{n}(y_i - \overline{y})^2}{n}} \tag{3}$$

$$\text{MRE} = \frac{1}{n}\sum_{i=1}^{n}\frac{|\hat{y}_i - y_i|}{y_i} \times 100\% \tag{4}$$

In the formula, $\hat{y}_i$ is model prediction; $y_i$ is actual sample value; $\overline{y}$ is average; and n is number of samples.

(2)   Significance test

Concerning the autocorrelation coefficient test table, when the degree of freedom (i.e., sample size) is 48 and the correlation coefficient is greater than 0.361, it reaches an extremely significant correlation level ($p < 0.01$). When the degree of freedom is 32 and the correlation coefficient value is greater than 0.436, it reaches an extremely significant correlation level ($p < 0.01$). When the degree of freedom is 16 and the correlation coefficient is greater than 0.590, it reaches an extremely significant correlation level ($p < 0.01$).

## 3. Results and Analysis

*3.1. Spectral Index Construction and Optimal Spectral Index Band Combination Extraction*

To maximize the utilization of information within hyperspectral reflectance data, this study selected 10 representative spectral indices. Firstly, the spectral indices of all bands of hyperspectral reflectance after 0–2-order fractional differential treatment were calculated by

band-by-band spectral indices. Then, the correlation matrix method was used to analyze the correlation between spectral indices and SMC. The i and j wavelengths with the largest correlation coefficient were used to construct different-order spectral indices. From the set of ten spectral indices, five with the strongest correlation to SMC were chosen to form the optimal spectral index combination. The correlation matrix diagram, depicted below, illustrates the correlation between the spectral indexes and SMC, ranging from a high negative correlation in blue to a high positive correlation in red.

Figures 3–7 show the correlation matrix diagrams of 0–2-order fractional differential hyperspectral spectral indexes and SMC after fractional differential treatment. The correlation of SMC with each order spectral index is greater than 0.361 ($p < 0.01$), reaching a very significant correlation level, indicating that the 10 spectral indexes selected in this study can be used to predict SMC at the flowering stage. The average values of the correlation coefficients between the 0–2-order spectral index and SMC were 0.523, 0.688, 0.728, 0.721, and 0.718, respectively. The correlation between the optimal spectral index and SMC calculated by the fractional differential treatment was significantly improved compared with the original spectral index. Under the 1.5-order differential treatment, the highest correlation coefficient with SMC was mNDI, the correlation coefficient value was 0.763, and the wavelength combination coordinate was (740, 696). The correlation coefficient values of each spectral index and SMC ranked from high to low are mNDI > TBI-1 > NDVI > TVI > TBI-3 > DI = SAVI = mSR > RI > TBI-2. From the aforementioned ten spectral indices, five, namely mNDI, TBI-1, NDVI, TVI, and TBI-3, exhibiting the highest correlation coefficients, were chosen to compose the optimal spectral index combinations. The associated bands, (740, 696), (726, 700), (688, 729), (754, 708), and (726, 700), were identified as the optimal spectral index band combinations. Table 4 displays the corresponding bands for both the remaining fractional differential optimal spectral index combination and the optimal spectral index combination.

### 3.2. Construction and Comparison of Soil Moisture Content Prediction Model

The optimal spectral index combination of each order was used as the independent variable, and SMC was used as the response variable. SVM, RF, and BPNN were used to construct the SMC estimation model of the soybean flowering stage. The accuracy of the model was comprehensively evaluated in terms of three aspects, namely $R^2$, RMSE, and MRE. The prediction results of different modeling methods for SMC are shown in Table 5. The results show that the $R^2$ of each SMC estimation model under different order differential transformation treatments is 1.5 order > 2 order > 1 order > 0.5 order > 0 order. The MRE performance of each model is 1.5 order < 2 order < 1 order < 0.5 order < 0 order. The RMSEs of the validation set of each model of 1.5 order are 0.01977, 0.00507, and 0.00579, which are smaller than the corresponding models of other orders. The validation set $R^2$ of the SVM, RF, and BPNN prediction models of SMC constructed by a 1.5-order differential spectral index are 0.90576, 0.91233, and 0.87778, respectively, which are higher than 0.539, reaching a very significant correlation level and having an excellent linear fitting effect. Under the same order differential transformation processing, the accuracy of the modeling set and verification set of the SMC estimation model constructed by the three modeling methods is as follows: RF > SVM > BPNN. Under each order differential treatment, the validation set $R^2$ of the SMC prediction model based on RF was 0.025–0.287 higher than that of SVM and GA-BP. The MRE decreased by 11.17–45.81%. In summary, the 1.5-order differential treatment and the RF model are the optimal differential order and the optimal model in this study, respectively. The $R^2$ of the modeling set and the verification set of the optimal SMC estimation model are 0.912 and 0.792, and the RMSEs are 0.912 and 0.792, respectively.

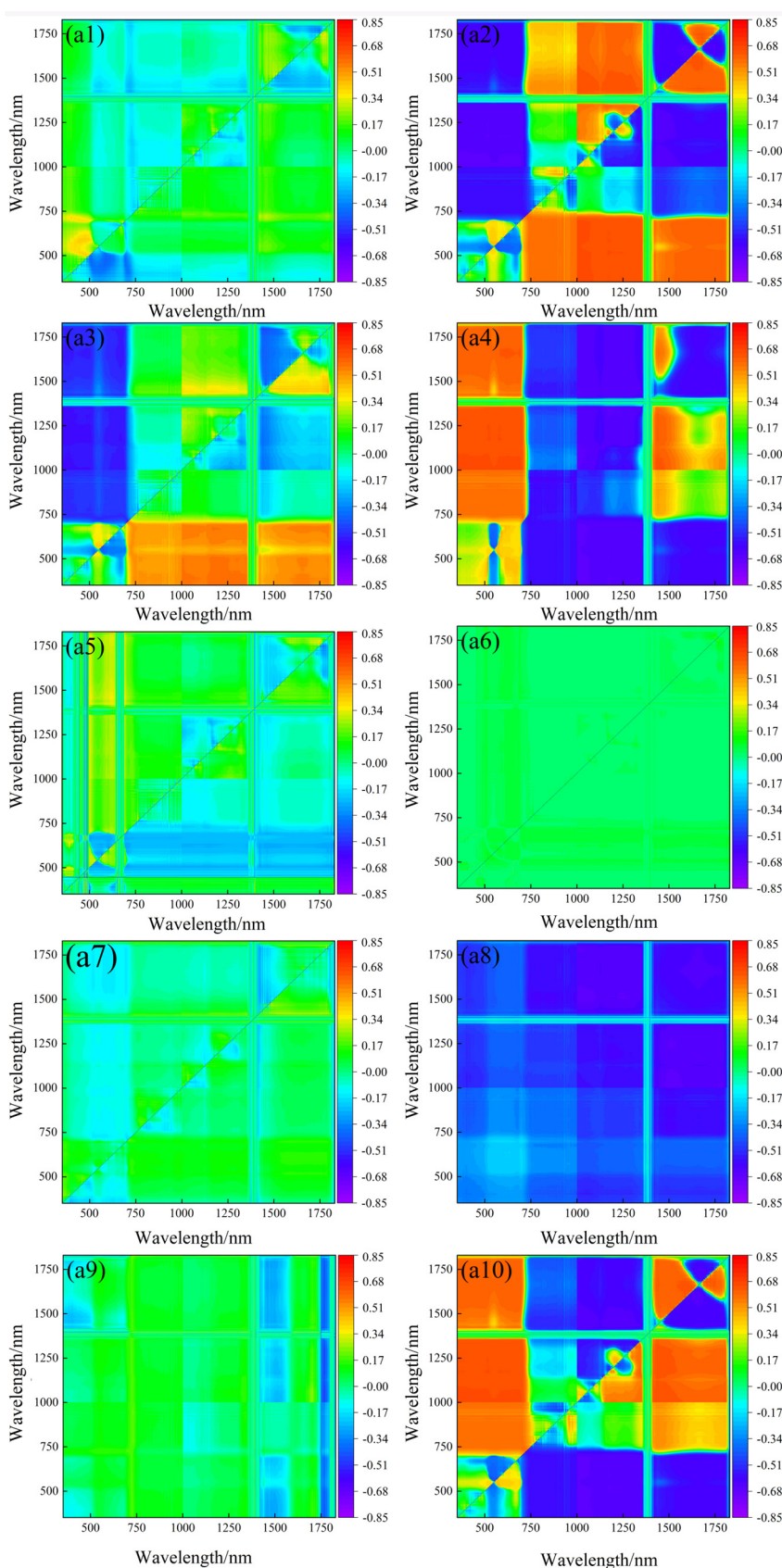

**Figure 3.** (**a1**–**a10**): The correlation matrix of RI, DI, SAVI, TVI, mSR, mNDI, NDVI, TBI-1, TBI-2, TBI-3, and soil moisture content under 0—order differential.

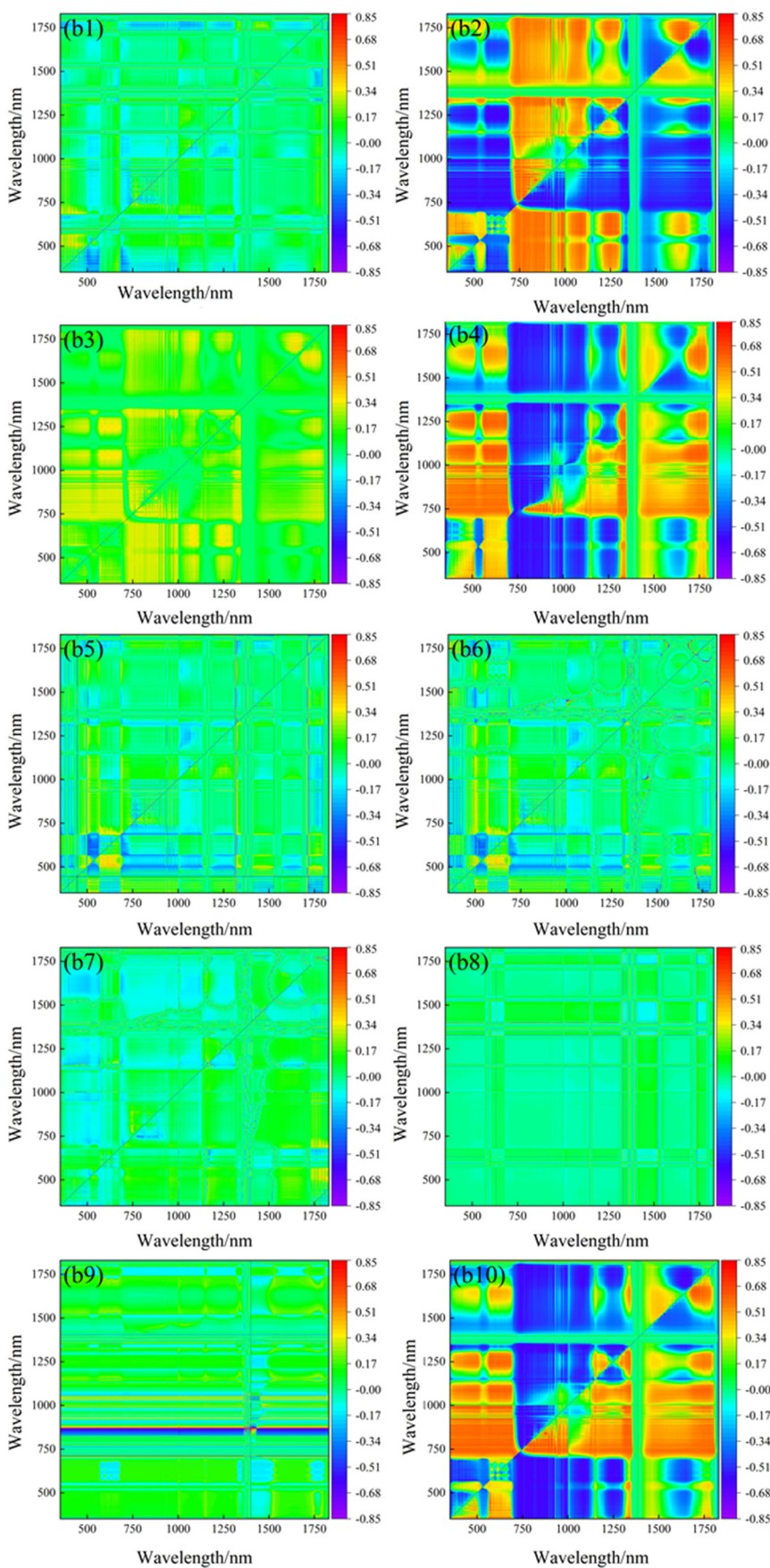

**Figure 4.** (**b1**–**b10**): The correlation matrix of RI, DI, SAVI, TVI, mSR, mNDI, NDVI, TBI-1, TBI-2, TBI-3, and soil moisture content under 0.5—order differential.

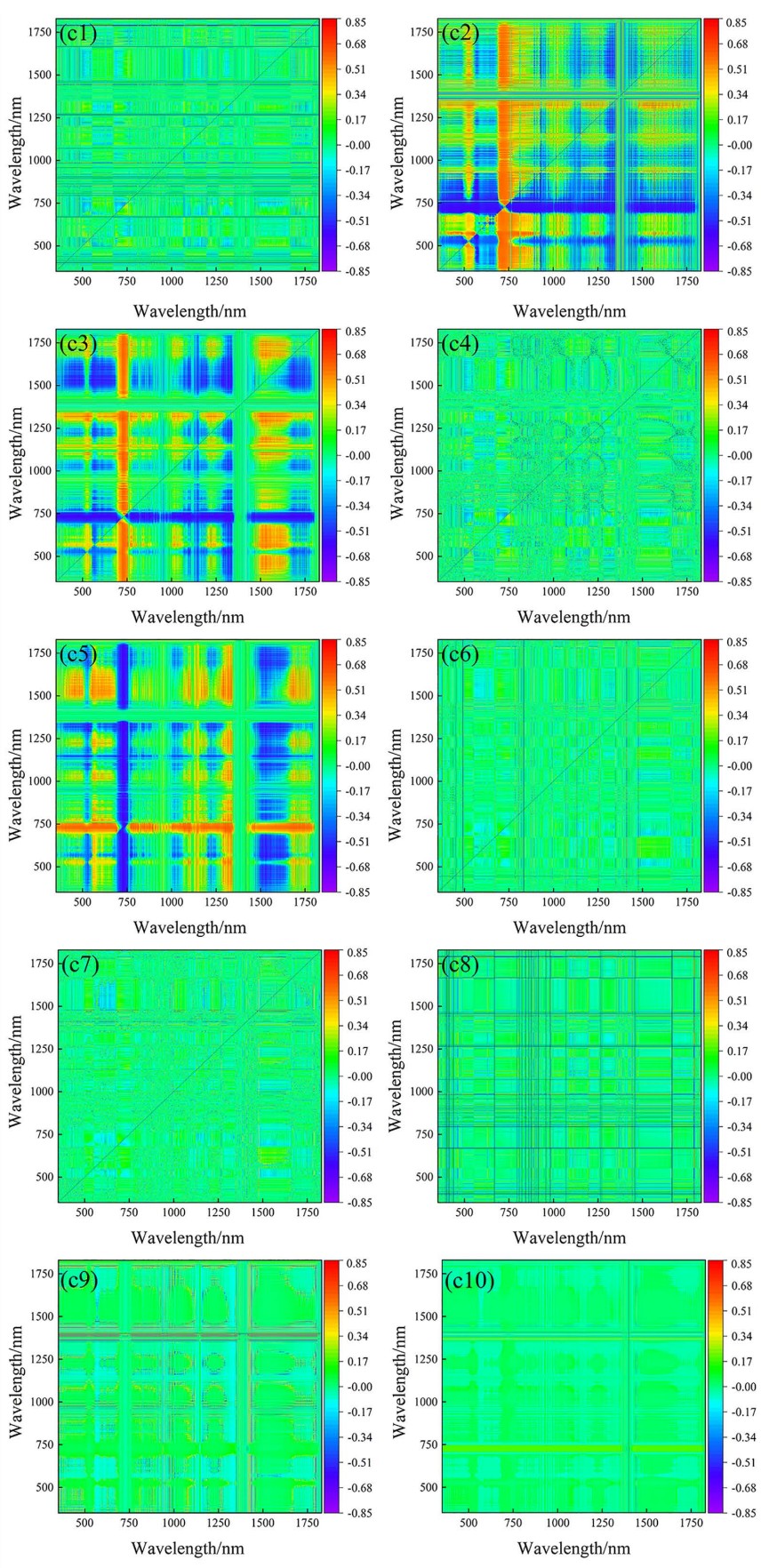

**Figure 5.** (**c1**–**c10**): The correlation matrix of RI, DI, SAVI, TVI, mSR, mNDI, NDVI, TBI-1, TBI-2, TBI-3, and soil moisture content under 1—order differential.

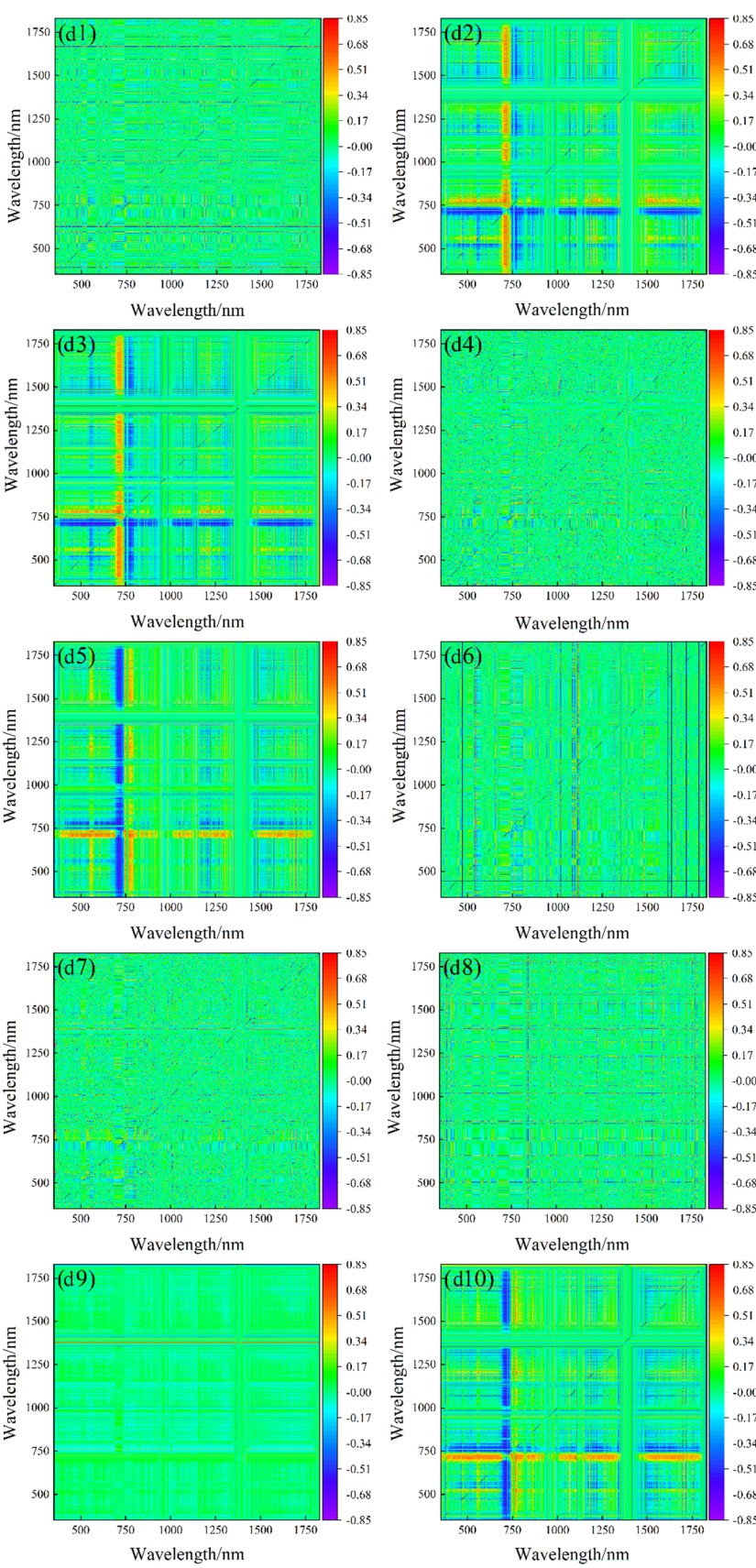

**Figure 6.** (**d1**–**d10**): The correlation matrix of RI, DI, SAVI, TVI, mSR, mNDI, NDVI, TBI-1, TBI-2, TBI-3, and soil moisture content under 1.5—order differential.

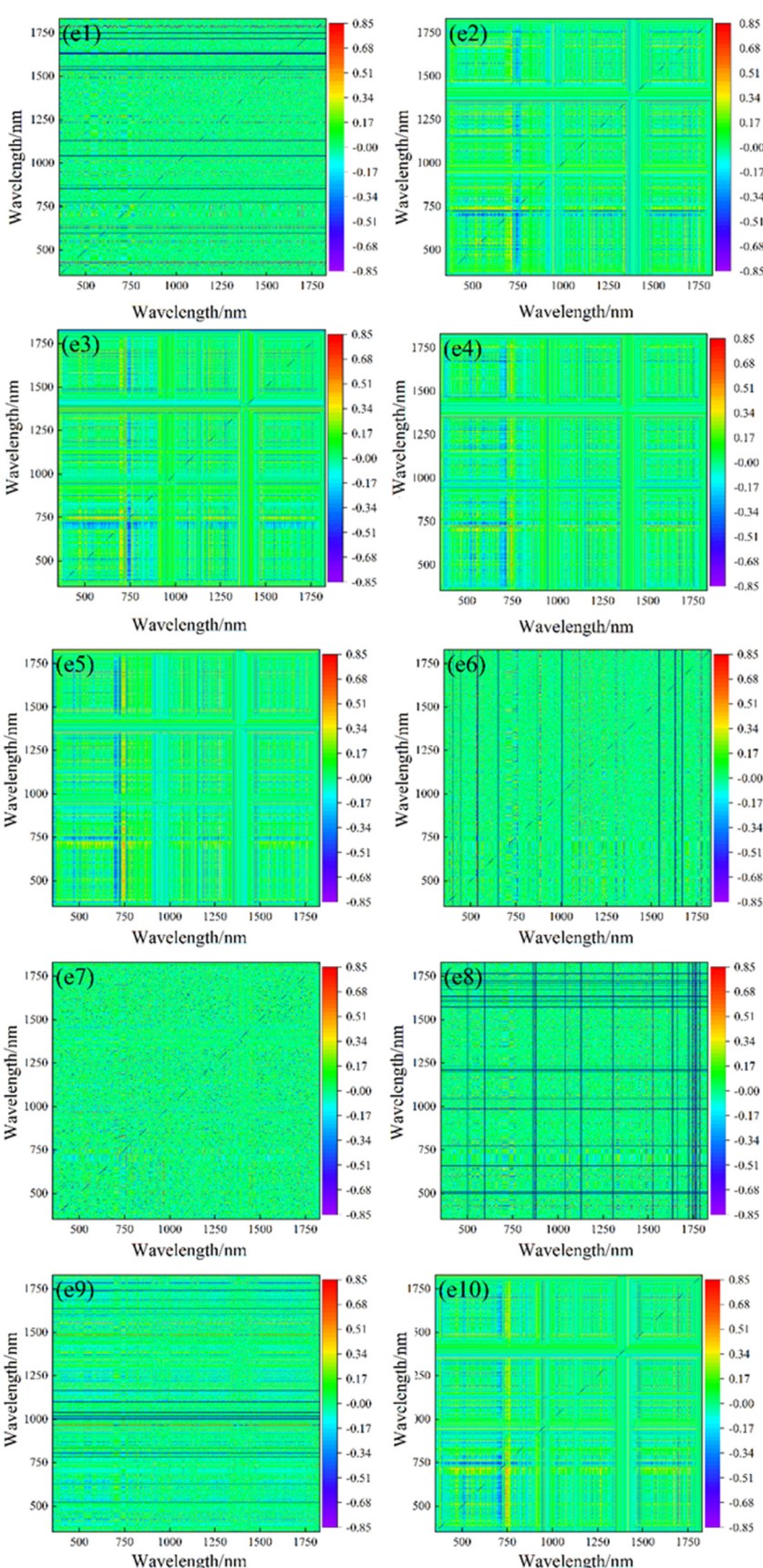

**Figure 7.** (**e1–e10**): The correlation matrix of RI, DI, SAVI, TVI, mSR, mNDI, NDVI, TBI-1, TBI-2, TBI-3, and soil moisture content under 2—order differential.

**Table 4.** Preferred spectral index wavelength combinations under various differential orders.

| Differential Order | Spectral Index | Correlation Coefficient | Position of Wavelength (i, j)/(nm) | Optimal Spectral Index Combination |
|---|---|---|---|---|
| 0 | DI | 0.647 ** | 747, 745 | DI, SAVI, TVI, mSR, TBI-3 |
|  | RI | 0.411 ** | 717, 720 |  |
|  | NDVI | 0.412 ** | 717, 720 |  |
|  | SAVI | 0.589 ** | 717, 720 |  |
|  | TVI | 0.659 ** | 759, 670 |  |
|  | mSR | 0.619 ** | 719, 718 |  |
|  | mNDI | 0.447 ** | 719, 718 |  |
|  | TBI-1 | 0.376 ** | 758, 759 |  |
|  | TBI-2 | 0.368 ** | 757, 717 |  |
|  | TBI-3 | 0.711 ** | 747, 745 |  |
| 0.5 | DI | 0.726 ** | 690, 720 | RI, SAVI, TBI-3, mNDI, NDVI |
|  | RI | 0.651 ** | 737, 748 |  |
|  | NDVI | 0.698 ** | 745, 737 |  |
|  | SAVI | 0.731 ** | 748, 737 |  |
|  | TVI | 0.682 ** | 757, 692 |  |
|  | mSR | 0.653 ** | 719, 717 |  |
|  | mNDI | 0.690 ** | 717, 720 |  |
|  | TBI-1 | 0.648 ** | 720, 753 |  |
|  | TBI-2 | 0.668 ** | 753, 696 |  |
|  | TBI-3 | 0.737 ** | 737, 748 |  |
| 1 | DI | 0.719 ** | 731, 721 | DI, NDVI, TBI-3, mNDI, TVI |
|  | RI | 0.688 ** | 673, 726 |  |
|  | NDVI | 0.722 ** | 711, 734 |  |
|  | SAVI | 0.716 ** | 726, 673 |  |
|  | TVI | 0.722 ** | 737, 696 |  |
|  | mSR | 0.713 ** | 721, 720 |  |
|  | mNDI | 0.763 ** | 757, 681 |  |
|  | TBI-1 | 0.700 ** | 729, 690 |  |
|  | TBI-2 | 0.628 ** | 700, 724 |  |
|  | TBI-3 | 0.719 ** | 673, 726 |  |
| 1.5 | DI | 0.721 ** | 726, 700 | mNDI, TVI, TBI-1, NDVI, TBI-3 |
|  | RI | 0.689 ** | 676, 738 |  |
|  | NDVI | 0.736 ** | 688, 729 |  |
|  | SAVI | 0.721 ** | 726, 700 |  |
|  | TVI | 0.726 ** | 754, 708 |  |
|  | mSR | 0.721 ** | 692, 726 |  |
|  | mNDI | 0.766 ** | 740, 696 |  |
|  | TBI-1 | 0.737 ** | 726, 700 |  |
|  | TBI-2 | 0.671 ** | 726, 677 |  |
|  | TBI-3 | 0.722 ** | 726, 700 |  |
| 2 | DI | 0.685 ** | 700, 726 | TVI, NDVI, TBI-1, mSR, mNDI |
|  | RI | 0.669 ** | 694, 746 |  |
|  | NDVI | 0.729 ** | 727, 682 |  |
|  | SAVI | 0.685 ** | 726, 700 |  |
|  | TVI | 0.703 ** | 723, 714 |  |
|  | mSR | 0.718 ** | 700, 726 |  |
|  | mNDI | 0.762 ** | 755, 720 |  |
|  | TBI-1 | 0.736 ** | 759, 694 |  |
|  | TBI-2 | 0.698 ** | 675, 676 |  |
|  | TBI-3 | 0.687 ** | 747, 737 |  |

'**' indicates that the correlation is significant at 0.01.

**Table 5.** Comparison of model prediction accuracy evaluation under different differential order.

| Differential Order | Evaluating Indicator | BPNN | | RF | | SVM | |
|---|---|---|---|---|---|---|---|
| | | Training Sets | Validation Set | Training Sets | Validation Sets | Training Set | Validation Set |
| 0 | $R^2$ | 0.629 | 0.652 | 0.714 | 0.722 | 0.642 | 0.679 |
| | RMSE (g/kg) | 0.007 | 0.027 | 0.005 | 0.006 | 0.006 | 0.006 |
| | MRE (%) | 4.31 | 4.477 | 3.467 | 4.622 | 3.903 | 4.565 |
| 0.5 | $R^2$ | 0.652 | 0.693 | 0.838 | 0.725 | 0.678 | 0.695 |
| | RMSE (g/kg) | 0.007 | 0.031 | 0.005 | 0.005 | 0.008 | 0.007 |
| | MRE (%) | 3.42 | 11.76 | 3.765 | 3.116 | 4.552 | 4.887 |
| 1 | $R^2$ | 0.794 | 0.707 | 0.842 | 0.737 | 0.799 | 0.719 |
| | RMSE (g/kg) | 0.007 | 0.029 | 0.006 | 0.007 | 0.02 | 0.029 |
| | MRE (%) | 4.428 | 5.471 | 3.164 | 4.831 | 4.444 | 6.4 |
| 1.5 | $R^2$ | 0.878 | 0.759 | 0.912 | 0.792 | 0.906 | 0.772 |
| | RMSE (g/kg) | 0.006 | 0.027 | 0.005 | 0.004 | 0.02 | 0.027 |
| | MRE (%) | 3.674 | 3.176 | 2.891 | 2.780 | 2.988 | 3.974 |
| 2 | $R^2$ | 0.863 | 0.745 | 0.889 | 0.762 | 0.867 | 0.757 |
| | RMSE (g/kg) | 0.02 | 0.007 | 0.006 | 0.004 | 0.02 | 0.026 |
| | MRE (%) | 3.918 | 5.189 | 2.448 | 2.542 | 2.838 | 5.574 |

## 4. Discussion

In this study, the differential transform method is introduced for processing vegetation canopy spectra, providing an advantageous technique for analyzing reflectance spectra. This approach effectively addresses the challenge of multicollinearity inherent in high-dimensional spectral data [47,48]. The employment of differential transform technology significantly impacts peak extraction in finely detailed spectra, thereby enhancing sensitivity to spectral features and curves. This method facilitates baseline correction [49,50], thereby intensifying the correlation between hyperspectral reflectance and SMC, subsequently enhancing inversion model accuracy [32,51]. As depicted in Figure 8, the highest accuracy is observed with 1.5-order processing. Consequently, the fractional order differential algorithm demonstrates superior capability in extracting SMC-relevant spectral data from hyperspectral spectrometers when compared to integer-order derivatives. Nevertheless, with increasing differential orders, background noise gradually diminishes while high-frequency noise progressively amplifies. This phenomenon concurrently leads to a reduction in potential sensitive information within reflectivity data, consequently lowering the signal-to-noise ratio of spectral information, thereby affecting model accuracy [52]. Table 4 displays the model results, indicating that the accuracy of certain fractional differential models surpasses that of raw spectral data, as well as first and second derivatives. Notably, accurate prediction models often cannot be established solely with raw spectral reflectance data. This underscores the foundational rationale behind employing spectral preprocessing for robust spectral data analysis.

The optimal band combination algorithm effectively addresses wavelength interactions within band combinations and handles the overlapping absorption of soil components [53]. This method has found application in numerous studies [54,55]. In the context of this study, five optimal spectral indices were selected from a pool of ten spectral indices. The varying maximum $R^2$ values derived from these optimal spectral indices under different fractional order differential transformations (Table 4) reveal dissimilar correlations between SMC and the spectral indexes. This disparity indicates variations in the ten spectral indices' propensity to correlate with SMC. Within the 1.5-order reflectivity context, the mNDI index excels, exhibiting a peak $R^2$ value of 0.766. Selecting the ideal spectral index and processing all spectral data are typically formidable tasks [23]. This underscores the advantage of spectral index combinations. Additionally, the methodology enhances modeling accuracy by extracting information-rich bands while eliminating irrelevant predictors, a contrast to full-spectrum data. The optimal spectral indices, constructed through band screening across the entire spectrum, encapsulate more meaningful information linked to SMC. Among the 50 spectral indices spanning 0–2 orders, 25 corresponding bands are positioned within the

red or near-infrared domains. As SMC considerably influences chlorophyll content, canopy chlorophyll indirectly reflects SMC [11]. The red edge band exhibits robust chlorophyll absorption and leaf reflection into the near-infrared, thus rendering it the fastest-growing region in green plant reflectance and a pivotal indicator of plant physiological traits [52]. The red edge band accommodates over 80% of plant physiochemical parameters' spectral information [56]. Hierarchical differential processing not only filters background noise but also preserves the red edge band's capacity to describe plant physiochemical parameters. As such, the band combinations sieved under each differential order in this study demonstrate a substantial correlation with SMC, primarily inhabiting the 670–760 nm range, corroborating prior research outcomes [57].

In this study, the optimal spectral index combinations for each order were chosen as input data, and three machine learning methods—SVM, RF and BPNN—were employed to construct SMC prediction models. Among these methods, the RF-based SMC prediction model exhibited the highest accuracy. This outcome underscores RF's robust capability to extract canopy chlorophyll-associated information from spectral reflectance data, consequently enhancing SMC inversion accuracy. The robustness of the RF algorithm, marked by its strong anti-interference and anti-overfitting attributes, alongside its high tolerance to background noise and outliers, renders it particularly adept at addressing nonlinear problems [58]. This conclusion aligns with the findings of Eyo et al. in their SMC inversion study [59]. BPNN stands as one of the most extensively employed neural network architectures. Through iterative adjustments of the network's interneuron weights, the algorithm minimizes the discrepancy between final output and anticipated results [60]. Yet, the inherent limitations of the BPNN algorithm, such as convergence into local extremes, weight converging to local minima, and sluggish convergence speed, tend to impede the accuracy and generalization capacity of neural network models [42]. In this study, the accuracy of the BPNN model was inferior to that of RF. This discrepancy might have arisen from the relatively limited sample size and the extensive model training iterations, contributing to a reduction in model precision and generalization capability [44]. When compared to RF and BPNN, the predictive efficacy of the SVM model on SMC was less impressive. This could potentially be attributed to SVM's limited anti-interference capacity and constraints stemming from parameter selection, including kernel functions and penalty factors [61].

Currently, substantial progress has been achieved in developing models for crop attributes such as leaf area index (LAI), biomass, nitrogen content, and chlorophyll content employing hyperspectral data [62–65]. However, the commonly employed ground-based hyperspectral data can solely be collected at specific locations, limiting their widespread application. The findings of this investigation reveal that optimal accuracy can be attained by utilizing a 1.5-order optimal spectral index combination as input variables and employing RF to construct a soil moisture content (SMC) prediction model. These findings necessitate validation and refinement through experimentation across varied regions, scales, and crop varieties. Such extensive validation ensures the model's adaptability and estimation precision, serving as a foundation for SMC prediction through diverse remote sensing techniques, including multispectral and UAV hyperspectral data. Furthermore, there exists an avenue for exploring fractional differential transformation of hyperspectral reflectance.

This study employed a step size of 0.5 for fractional order differentiation, which may result in relatively lower accuracy in processing high-spectral information and weaker details in data handling compared to a smaller step size, with a relatively modest compensatory effect. Additionally, the selection of a fixed band in the construction of the three-dimensional spectral index had a certain impact on modeling accuracy. This step lays the groundwork for subsequent fractional order differentiation with a smaller step size. Moreover, this study provides a scientific basis for the rapid and accurate determination of soil moisture content during the crop growing season.

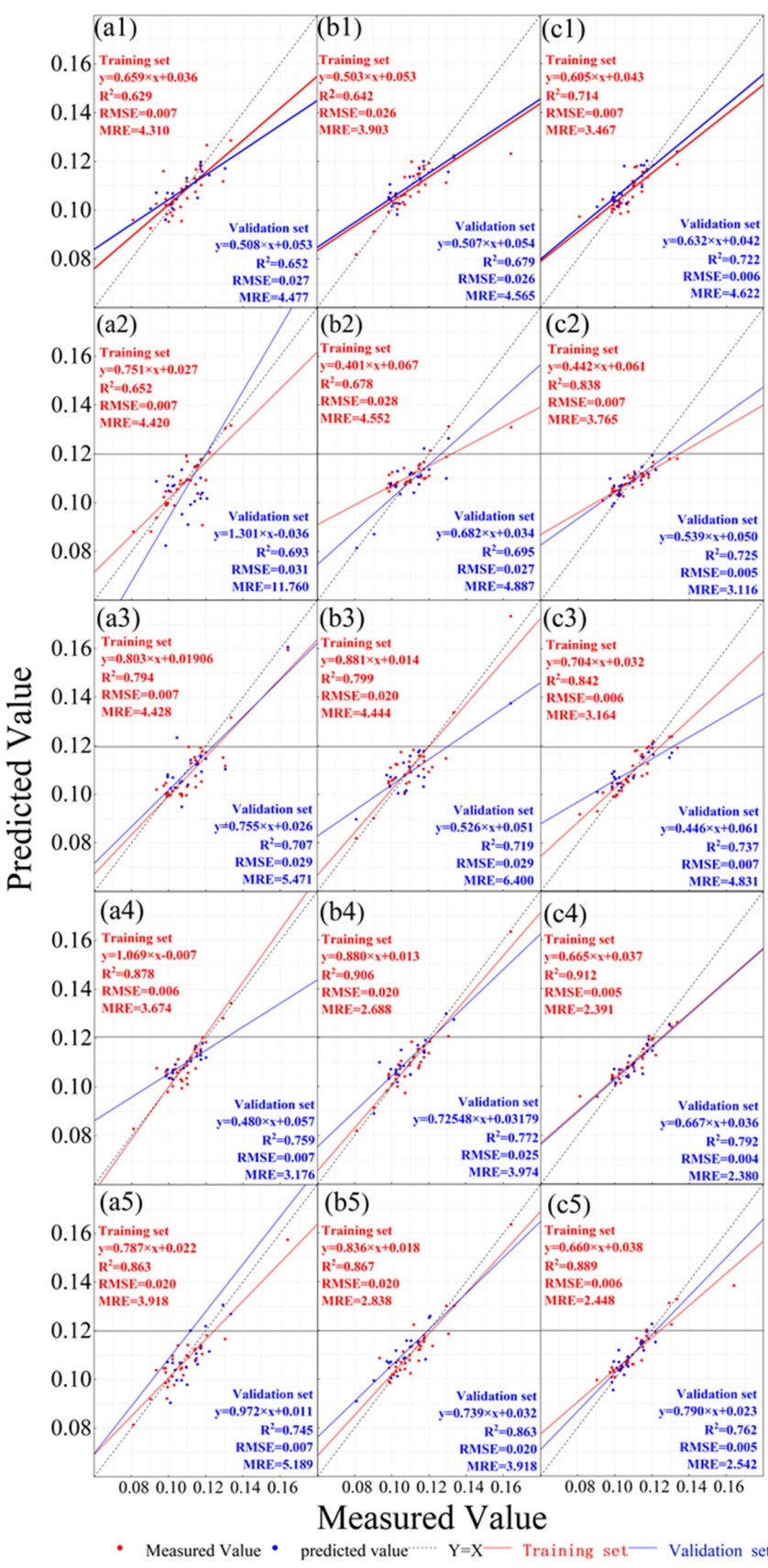

**Figure 8.** Model evaluation results. (**a1**–**a5**) is the 0–2−order soil moisture content prediction model constructed by the BPNN method; (**b1**–**b5**) are the results of the 0–2−order soil moisture content prediction model constructed by the SVM method. (**c1**–**c5**) are the results of the 0–2−order soil moisture content prediction model constructed by the RF method.

### 5. Conclusions

In this study, the SMC at the flowering stage was taken as the research object, and the SMC and canopy hyperspectral reflectance data were measured. The original hyperspectral reflectance data were processed by 0–2-order fractional differential, and the band-by-band spectral index was calculated. The correlation matrix method was used to extract the optimal wavelength combination for different order spectral index construction. Finally, based on the optimal spectral index of each order and three machine learning models, namely SVM, RF and BPNN, the SMC prediction model was constructed. The conclusions are as follows:

(1) Compared with the original hyperspectral reflectance data, the correlation between the optimal spectral index of each order extracted after fractional differential transformation and SMC was significantly improved. The average value of the correlation coefficient between each spectral index and SMC under the 1.5-order treatment was 0.380% higher than that of the original spectral index. Among them, mNDI showed the highest correlation, with a correlation coefficient of 0.766.

(2) When the modeling method is the same, and the input variables are different, the accuracy of the SMC prediction model is as follows: 1.5 order > 2.0 order > 1.0 order > 0.5 order > original order. When the input variables are the same and the modeling method changes, the accuracy of the SMC prediction model is as follows: RF > SVM > BPNN. A comprehensive comparison of the model's evaluation indicators shows that the 1.5-order differential and RF methods are the optimal differential order and optimal model construction methods in this study, respectively. The $R^2$ of the optimal SMC estimation model modeling set and validation set are 0.912 and 0.792, the RMSE is 0.00507 and 0.00393, and the MRE is 2.3901% and 2.3802%.

**Author Contributions:** W.L.: Methodology, Formal analysis, Writing—original draft, Visualization. X.L. and Z.T.: Methodology, Software. X.W. and X.H.: Data curation, Writing—review and editing, Formal analysis. Y.X.: Supervision, Writing—review and editing, Funding acquisition. M.C., Y.D., L.M., S.W., H.S., Y.Z. and Z.L.: Data curation. F.Z.: Supervision, Project administration. All authors have read and agreed to the published version of the manuscript.

**Funding:** This work was supported by the National Natural Science Foundation of China: No. 52179045.

**Data Availability Statement:** Data are contained within the article.

**Conflicts of Interest:** The authors declare that they have no known competing financial interests or personal relationships that could have appeared to influence the work reported in this paper.

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
