# Peer review of "Estimation of Soil Moisture Content Based on Fractional Differential and Optimal Spectral Index"

_agronomy, doi:10.3390/agronomy14010184_

Round 1
Reviewer 1 Report
Comments and Suggestions for Authors
In this paper, the authors tackled an important problem of estimating the soil moisture from remotely sensed hyperspectral image data. The topic is certainly worthy of investigation and easily falls into the scope of the journal. However, there are quite a number of important issues which should be addressed thoroughly before the manuscript could be, in my opinion, considered for publication:
1. Although the English is acceptable, the manuscript would still benefit from careful proofreading, perhaps with a help of a native speaking colleague. There are quite a number of vague statements around the manuscript (even in the abstract), as well as grammar errors. Also, please avoid using short forms, such as “it’s”.
2. The authors did review the state of the art in analyzing soil from highly-dimensional image data. There is, however, an important advantage of such remotely sensed data, being its extreme scalability. It would be useful to contextualize the work reported here in on-board machine learning, especially given that the proposed framework is fairly lightweight and could be potentially deployed on board imaging satellites (see e.g., the work by Wijata: https://ieeexplore.ieee.org/document/10180082).
3. All entities in the manuscript, such as figures and tables, should be self-contained and possible to understand without diving into the text. To this end, please expand the captions of the figures – as an example, Figure 1 is fairly challenging to analyze without diving into the corresponding text. Similarly, the table captions should be expanded as well.
4. It would be useful to provide a schematic (visual) view of the experimental design discussed in Section 2.1.
5. All entities in the paper (again, figures, tables, and so forth) should be referenced from the text.
6. Table 1 is quite vague – what do the authors mean by saying that the “soil moisture content” is 48 for “sample size”?
7. Please carefully review the symbols and equations (see e.g., lines 190-191).
8. The authors claim that the exploited spectral indices are “optimal” – please formally prove this “optimality”.
9. The quality of the figures should be improved – all of them should be high-resolution in a vector format.
10. In Table 4, the authors report different results for two “training sets” for random forests. I presume the second “training” test is actually the validation set – please revise carefully.
11. How were the hyperparameters of the machine learning models fine-tuned? Please discuss in detail.
12. Currently, we are facing the reproducibility crisis in the machine learning field. To address it, the authors should make their implementation of the proposed method publicly available. The same applies to the dataset, which should be made public (or at least a sensible subset of it should be public, so that other research groups can easily reproduce the experiments).
13. Are the differences across the investigated algorithms statistically significant? Please report appropriate p-values.
14. It would be useful to perform a bit more rigorous validation of the proposed techniques, following the cross-validation settings.
Comments on the Quality of English LanguageThe manuscript would benefit from proofreading.
Author Response
Responses to Reviewer #1 (Manuscript ID: Agronomy-2799647)
Thank you for addressing all my comments.
Thank you for your careful review and positive comments. We have now incorporated the reviewer’s comments and suggestions in preparation of the revised manuscript. The modified part is marked in red in the manuscript.
- Although the English is acceptable, the manuscript would still benefit from careful proofreading, perhaps with a help of a native speaking colleague. There are quite a number of vague statements around the manuscript (even in the abstract), as well as grammar errors. Also, please avoid using short forms, such as “it’s”.
A: Thank you for pointing this out. The detailed content has been modified. Please refer to the manuscript file for review.
- The authors did review the state of the art in analyzing soil from highly-dimensional image data. There is, however, an important advantage of such remotely sensed data, being its extreme scalability. It would be useful to contextualize the work reported here in on-board machine learning, especially given that the proposed framework is fairly lightweight and could be potentially deployed on board imaging satellites (see e.g., the work by Wijata: https://ieeexplore.ieee.org/document/10180082).
A: Thank you for pointing this out. This study has been revised as suggested
Line118-119: This also provides a theoretical basis for artificial intelligence applications to quickly and accurately analyze large-scale hyperspectral satellite remote sensing information
- All entities in the manuscript, such as figures and tables, should be self-contained and possible to understand without diving into the text. To this end, please expand the captions of the figures – as an example, Figure 1 is fairly challenging to analyze without diving into the corresponding text. Similarly, the table captions should be expanded as well.
A: Thank you for pointing this out. The detailed content has been modified. Please refer to the manuscript file for review.
Line132: Figure 1. Study area.
- It would be useful to provide a schematic (visual) view of the experimental design discussed in Section 2.1.
A: Thank you for pointing this out. The experimental design scheme has been added to Section 2.1
Line149: Table 1Test scheme design. The detailed content has been modified. Please refer to the manuscript file for review.
- All entities in the paper (again, figures, tables, and so forth) should be referenced from the text.
A: Thank you for pointing this out. The detailed content has been modified. Please refer to the manuscript file for review.
- Table 1 is quite vague – what do the authors mean by saying that the “soil moisture content” is 48 for “sample size”?
A: Thank you for pointing this out. It has been supplemented in the relevant parts.
Line165-172: In this study, six sites were randomly selected in each plot during the flowering period of soybean to determine the soil moisture content. The mean value of the six sampling points was the soil moisture content of the plot, a total of 24 plots, and the corresponding hyper-spectral remote sensing information was obtained at the same time. A total of 48 groups of samples were tested in two years. A total of 48 groups of SMC and hyperspectral reflec-tance samples were obtained in the two-year experiment(Table 2).
- Please carefully review the symbols and equations (see e.g., lines 190-191).
A: Thank you for pointing this out. The article has been revised
Line208:
- The authors claim that the exploited spectral indices are “optimal” – please formally prove this “optimality”.
A: Thank you for pointing this out. In this study, five spectral indices with the highest correlation were selected from 10 spectral indices as the optimal spectral indices.
- The quality of the figures should be improved – all of them should be high-resolution in a vector format.
A: Thank you for pointing this out. The article has been revised
- In Table 4, the authors report different results for two “training sets” for random forests. I presume the second “training” test is actually the validation set – please revise carefully.
A: Thank you for pointing this out. The article has been revised
Line342: Validation Sets
- How were the hyperparameters of the machine learning models fine-tuned? Please discuss in detail.
A: Thank you for pointing this out. The hyperparameters for SVM include the penalty factor and the radial basis function parameter. The penalty factor was tuned within the range of 0.1 to 100, and the radial basis function parameter was adjusted in the range of 0.001 to 10. After manual tuning, the optimal hyperparameters were determined as a penalty factor of 4 and a radial basis function parameter of 0.8. For RF, the hyperparameters involve the number of decision trees and the minimum leaf size. The number of decision trees was adjusted in the range of 50 to 500, and the minimum leaf size was tuned in the range of 1 to 10. During the training-testing process of the RF model, an exhaustive search was conducted across features and methods to effectively determine the optimal number of decision trees. After multiple rounds of training and error analysis, the RF model was finalized with 100 decision trees and a minimum leaf size of 5. BPNN's hyperparameters include the maximum number of iterations, error threshold, and learning rate. Through manual tuning, the optimal hyperparameters were determined as a maximum of 1000 iterations, an error threshold of 1×10^(-6), and a learning rate of 0.01.
- Currently, we are facing the reproducibility crisis in the machine learning field. To address it, the authors should make their implementation of the proposed method publicly available. The same applies to the dataset, which should be made public (or at least a sensible subset of it should be public, so that other research groups can easily reproduce the experiments).
A: Thank you for pointing this out. The detailed parameters of the machine learning algorithms will be shared with some data after the paper is included.
- Are the differences across the investigated algorithms statistically significant? Please report appropriate p-values.
|
0.5 Correlations |
||||
|
|
VAR00001 |
VAR00002 |
VAR00003 |
|
|
VAR00001 |
Pearson Correlation |
1 |
.881** |
.665** |
|
Sig. (2-tailed) |
|
.000 |
.000 |
|
|
N |
48 |
48 |
48 |
|
|
VAR00002 |
Pearson Correlation |
.881** |
1 |
.665** |
|
Sig. (2-tailed) |
.000 |
|
.000 |
|
|
N |
48 |
48 |
48 |
|
|
VAR00003 |
Pearson Correlation |
.665** |
.665** |
1 |
|
Sig. (2-tailed) |
.000 |
.000 |
|
|
|
N |
48 |
48 |
48 |
|
|
1.0 Correlations |
||||
|
|
VAR00001 |
VAR00002 |
VAR00003 |
|
|
VAR00001 |
Pearson Correlation |
1 |
.734** |
.818** |
|
Sig. (2-tailed) |
|
.000 |
.000 |
|
|
N |
48 |
48 |
48 |
|
|
VAR00002 |
Pearson Correlation |
.734** |
1 |
.731** |
|
Sig. (2-tailed) |
.000 |
|
.000 |
|
|
N |
48 |
48 |
48 |
|
|
VAR00003 |
Pearson Correlation |
.818** |
.731** |
1 |
|
Sig. (2-tailed) |
.000 |
.000 |
|
|
|
N |
48 |
48 |
48 |
|
|
1.5 Correlations |
||||
|
|
VAR00001 |
VAR00002 |
VAR00003 |
|
|
VAR00001 |
Pearson Correlation |
1 |
.829** |
.842** |
|
Sig. (2-tailed) |
|
.000 |
.000 |
|
|
N |
48 |
48 |
48 |
|
|
VAR00002 |
Pearson Correlation |
.829** |
1 |
.900** |
|
Sig. (2-tailed) |
.000 |
|
.000 |
|
|
N |
48 |
48 |
48 |
|
|
VAR00003 |
Pearson Correlation |
.842** |
.900** |
1 |
|
Sig. (2-tailed) |
.000 |
.000 |
|
|
|
N |
48 |
48 |
48 |
|
|
2.0 Correlations |
||||
|
|
VAR00001 |
VAR00002 |
VAR00003 |
|
|
VAR00001 |
Pearson Correlation |
1 |
.764** |
.248 |
|
Sig. (2-tailed) |
|
.000 |
.089 |
|
|
N |
48 |
48 |
48 |
|
|
VAR00002 |
Pearson Correlation |
.764** |
1 |
.475** |
|
Sig. (2-tailed) |
.000 |
|
.001 |
|
|
N |
48 |
48 |
48 |
|
|
VAR00003 |
Pearson Correlation |
.248 |
.475** |
1 |
|
Sig. (2-tailed) |
.089 |
.001 |
|
|
|
N |
48 |
48 |
48 |
|
A: Thank you for pointing this out. From the table, it can be observed that, except for the low correlation between BPNN and RF in the 2.0 order, other machine learning models exhibit a significant correlation, indicating statistical significance.
- It would be useful to perform a bit more rigorous validation of the proposed techniques, following the cross-validation settings.
A: Thank you for pointing this out. We will use the cross-validation method for research verification in subsequent work.
Reviewer 2 Report
Comments and Suggestions for Authors
According to the title, a presentation of soil moisture estimation based on fractional differential and optimal spectral index is expected, however, this is not accompanied by results, discussion and conclusions.
There is no mention of the soil in the conclusion itself, and the emphasis throughout the paper is exclusively on the method.
There are also a number of necessary corrections such as:
Figure 1. kilometers is the correct term that should be used
Table 1 - why the number of samples is a decimal number; The CV cannot be 0.12% because it is the ratio of the standard deviation to the mean value and according to the displayed values it would be 12.5% in %
Table 2: what does "select index" mean?!
It is not explained why exactly the values 0-2 order soil moisture were chosen
The conclusion does not even mention the soil and its moisture.
My opinion is that some other journal would be more appropriate, or the authors should greatly adapt the work to the journal and to what is stated in the title.
Sincerely
Author Response
Responses to Reviewer #2 (Manuscript ID: Agronomy-2799647)
Thank you for addressing all my comments.
Thank you for your careful review and positive comments. We have now incorporated the reviewer’s comments and suggestions in preparation of the revised manuscript. The modified part is marked in red in the manuscript.
Figure 1. kilometers is the correct term that should be used
A: Thank you for pointing this out. The article has been revised
Table 1 - why the number of samples is a decimal number; The CV cannot be 0.12% because it is the ratio of the standard deviation to the mean value and according to the displayed values it would be 12.5% in %
A: Thank you for pointing this out. The article has been revised
Line172:12.5
Table 2: what does "select index" mean?!
It is not explained why exactly the values 0-2 order soil moisture were chosen
The conclusion does not even mention the soil and its moisture.
A: Thank you for pointing this out. Select index is based on previous studies, this study selected 10 spectral indices related to soil water content ,
In this study, 10 spectral indices were established, and 5 spectral indices with high correlation were selected. The main reason is to reduce the complexity of modeling and improve the accuracy of modeling.
XIANG Y.Z WANG X. AN J.Q TANG Z.J. LI W.Y. SHI H.Z. Estimation of Leaf Area lndex of Soybean Based on Fractional Order Differentiation and Optimal Spectral lndex
The purpose of this study is to provide a basis for rapid detection of soil water content.
Reviewer 3 Report
Comments and Suggestions for Authors
The presented research is actual and novel. Thanks to the rapid development of information technology, the use of hyperspectral data becomes more perspective in various fields of activity, including precision agriculture. The problem of accurately determining soil moisture content in each area of an agricultural field is of high importance given global climate changes. It would be advisable to further expand the scope of the study to other crops, growth phases and climatic conditions.
There are several comments:
1. It is recommended to expand the overview of existing approaches to determining soil moisture (sensors, laboratory samples, etc.) in the introduction.
2. In Fig. 1 it is not entirely clear where the NMO is located; if it is an asterisk, then it is necessary to provide explanations for the figure. In addition, the location of the test plots is not obvious.
3. In table. 1: if it is indicated that the coefficient of variation is presented as a percentage, then most likely the value should be 12.5%, and not 0.12.
4. It is recommended to bring Table 2 into uniformity: indicate wavelengths instead of i, j, or decipher what kind of bands are meant (red, near-infrared, etc.).
5. Line 257: "the correlation between SMC and SMC" is probably a typo.
6. Line 257: "The i and j wavelengths..." - it is recommended to clarify what kind of bands are used in i and j.
7. It is recommended to expand the description of the dataset (number of measurements used, etc.); it is possible to provide, as an example, a graph of spectral characteristics for several test plots.
8. It is recommended to add research limitations to the discussion or as a separate section.
Author Response
Responses to Reviewer #3 (Manuscript ID: Agronomy-2799647)
Thank you for addressing all my comments.
Thank you for your careful review and positive comments. We have now incorporated the reviewer’s comments and suggestions in preparation of the revised manuscript. The modified part is marked in red in the manuscript.
- It is recommended to expand the overview of existing approaches to determining soil moisture (sensors, laboratory samples, etc.) in the introduction.
A: Thank you for pointing this out. The detailed content has been modified. Please refer to the manuscript file for review.
- In Fig. 1 it is not entirely clear where the NMO is located; if it is an asterisk, then it is necessary to provide explanations for the figure. In addition, the location of the test plots is not obvious.
A: Thank you for pointing this out. The detailed content has been modified. Please refer to the manuscript file for review.
- In table. 1: if it is indicated that the coefficient of variation is presented as a percentage, then most likely the value should be 12.5%, and not 0.12.
A: Thank you for pointing this out. This study has been revised as suggested.
Line172:12.5
- It is recommended to bring Table 2 into uniformity: indicate wavelengths instead of i, j, or decipher what kind of bands are meant (red, near-infrared, etc.).
A: Thank you for pointing this out. In this study, the hyperspectral 350-1830 range, the results of the correlation analysis of the preferred band, the band is not fixed, so the use of i, j instead
- Line 257: "the correlation between SMC and SMC" is probably a typo.
A: Thank you for pointing this out. This study has been modified to the correlation between spectral indices and SMC.
Line276-277: Then the correlation matrix method was used to analyze the correlation between spectral indices and SMC.
- Line 257: "The i and j wavelengths..." - it is recommended to clarify what kind of bands are used in i and j.
A: Thank you for pointing this out. In this study, the hyperspectral 350-1830 range, the results of the correlation analysis of the preferred band, the band is not fixed, so the use of i, j instead
- It is recommended to expand the description of the dataset (number of measurements used, etc.); it is possible to provide, as an example, a graph of spectral characteristics for several test plots.
A: Thank you for pointing this out. This study has been supplemented in the relevant parts.
Line: This study selected the original hyperspectral information under SFMN3, SMN3, FMN3, and NMN3 treatments and plotted the spectral characteristic curves, as shown in Figure 2
- It is recommended to add research limitations to the discussion or as a separate section.
A: Thank you for pointing this out. This study has been supplemented in the relevant parts.
Line429-436: This study employed a step size of 0.5 for fractional-order differentiation, which may result in relatively lower accuracy in processing high-spectral information and weaker details in data handling compared to a smaller step size, with a relatively modest compensatory effect. Additionally, the selection of a fixed band in the construction of the three-dimensional spectral index had a certain impact on modeling accuracy. This step lays the groundwork for subsequent fractional-order differentiation with a smaller step size. Moreover, this study provides a scientific basis for the rapid and accurate determination of soil moisture content during the crop growing season
Round 2
Reviewer 1 Report
Comments and Suggestions for Authors
Thank you indeed for addressing most of my concerns. The authors should, however, do yet another pass through the entire manuscript---here are unresolved references and typos around the paper.
Comments on the Quality of English LanguageThe English is fine.
Reviewer 2 Report
Comments and Suggestions for Authors
all my suggestions and comments have been adopted and the results of the study are much clearer.
Now the work is at the level expected in this journal